Methods

# *wgbstools*: a computational suite for DNA methylation sequencing data analysis

Netanel Loyfer[1,2] , Jonathan Rosenski[1] , Tommy Kaplan[1,2,3,4]

**Next-generation methylation-aware sequencing of DNA sheds light on the fundamental role of methylation in cellular function in health and disease, increasing the number of covered CpG sites from hundreds of thousands in previous array-based approaches to tens of millions across the whole genome. While array-based approaches are limited to single-CpG resolution, next-generation sequencing allows for a more detailed, single-molecule fragment-level analysis; however, existing tools to fully use this capability are not yet well developed. Here, we present *wgbstools*, an extensive computational suite tailored for methylation sequencing data. *wgbstools* allows fast access and ultracompact anonymized representation of high-throughput methylome data, obtained through various library preparation and sequencing methods, with a custom epiread file format achieving a compression factor of over 100x from the input BAM file. In addition, *wgbstools* contains state-of-the-art algorithms for genomic segmentation, biomarker identification, genetic and epigenetic data integration, and more. *wgbstools* offers fragment-level analysis and informative visualizations, across multiple genomic regions and samples.**

## Introduction

DNA methylation is an essential component of gene regulation and genome packaging, and is a fundamental mark of cell identity. Indeed, alterations and dysregulation of DNA methylation are implicated in regulating critical processes of development, aging, and disease (1). Analysis of DNA methylation is therefore key for understanding these processes, as well as for clinical diagnosis. Recently, methylation-based analysis of circulating cell-free DNA fragments is emerging as an accurate tool for quantifying cell type–specific damage in clinical diagnosis and monitoring of various diseases, including several types of cancer (2, 3, 4, 5, 6, 7, 8).

Genomic studies of DNA methylation often use the Illumina BeadChip 450K/EPIC arrays, with EPIC v1 and v2 covering 850 and 950 K CpG sites, respectively, which measure the average DNA methylation levels (beta values) at a predefined set of CpG sites, including a mere 1.5–3% of the 28 million CpG methylation sites across the entire human genome (9, 10). These technologies enforce a limitation of analyzing methylation at each single CpG and are blind to the information of methylation status of neighboring, and likely similarly methylated, CpGs of each sequenced DNA read, reducing noise and allowing for the detection of allele-specific methylation (5, 11). Recent advances in DNA-sequencing technologies have facilitated a fragment-centered view that captures binary DNA methylation patterns of multiple neighboring CpG sites, at a single-molecule resolution, and even allows for the possibility to incorporate the genetic information of each sequenced molecule (4, 5, 6, 7, 8, 12, 13, 14, 15). These technologies include conversion methods such as bisulfite (16) or enzymatic methyl treatment followed by sequencing (EM-seq) (17), alongside direct detection of base modifications, using methylation-aware DNA-sequencing technologies such as Oxford Nanopore Technologies (ONT) (18) or PacBio (19). These technologies allow methylation to be measured across the entire genome, or enriched at target regions using hybrid capture arrays, restriction enzymes (RRBS), or targeted PCR (3, 4, 5, 20, 21, 22). Nonetheless, the computational and algorithmic tools for processing, visualizing, and analyzing methylation across fragments lag behind. Although several excellent tools exist for processing WGBS and RRBS data (23), *wgbstools* represents a shift in perspective, with genome segmentation that identifies CpG-resolution start and end sites marking methylation change-points, as well as a fragment-level analysis through bimodal methylation identification and homog command.

Here, we present *wgbstools* (github.com/nloyfer/wgbs_tools), an open-source computational suite for efficient end-to-end processing, data conversion, representation, anonymization, visualization, and analysis of sequencing DNA methylation data, across multiple sequencing platforms and library preparation methods.

[1]School of Computer Science and Engineering, The Hebrew University of Jerusalem, Jerusalem, Israel   [2]Faculty of Medicine, The Hebrew University of Jerusalem, Jerusalem, Israel   [3]Barts Cancer Institute, Queen Mary University of London, London, UK   [4]Centre for Epigenetics, Queen Mary University of London, London, UK

Correspondence: tommy.kaplan@mail.huji.ac.il

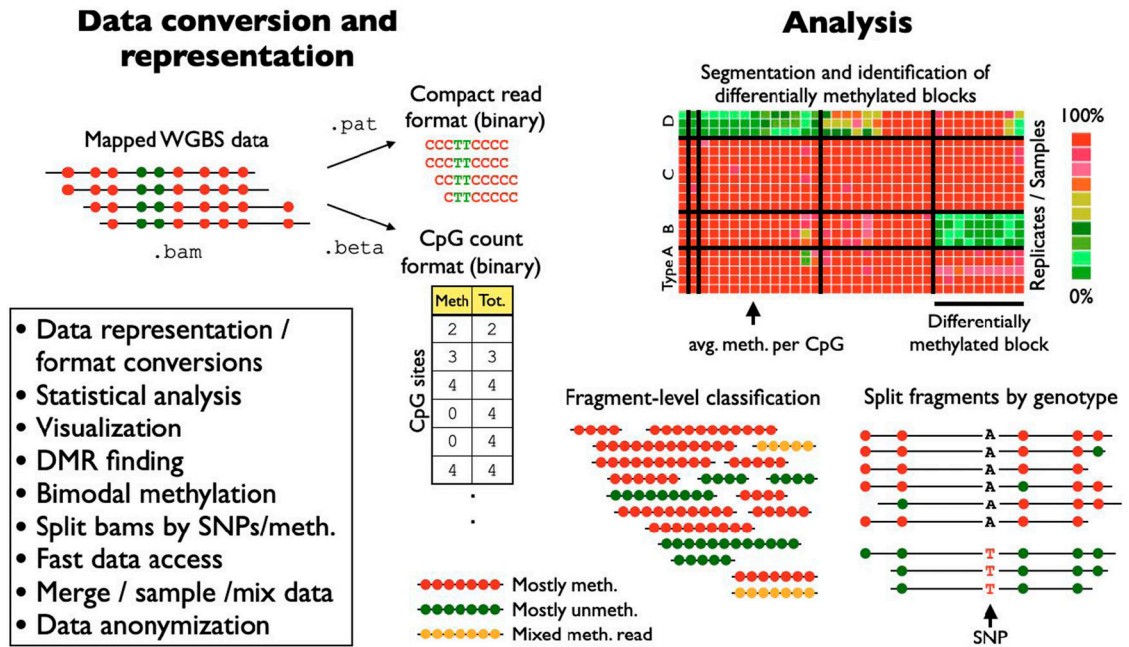

**Figure 1.  *wgbstools*: a software suite for DNA methylation data representation, analysis, and visualization.**
Given a BAM file containing next-generation sequencing of DNA methylation data, *wgbstools* converts mapped reads (or read pairs) to compact representations of methylation data at fragment-level resolution, thus maintaining methylation patterns across adjacent CpG sites from a single molecule. In addition, *wgbstools* provides a CpG-oriented count format, which includes the number of reads in which any CpG was methylated and sequenced. All *wgbstools* formats are fully indexed to support efficient direct access and include multiple visualization schemes, for command-line explorations and publication-quality figures. *wgbstools* contains several machine-learning algorithms for optimal segmentation of genome-scale data into continuous methylation blocks of dynamic size; for the identification of cell type–specific differentially methylated regions; and SNP-based allele-specific methylation data analysis.

# Results

### Data representation

The primary format used for DNA methylation sequencing data is the BAM file format, which encompasses sequenced reads, names, genomic mapping, quality, and read pair information (24). However, working with these files can be challenging because of their considerable size and complexity, making simultaneous analysis of multiple samples arduous. Furthermore, BAM files store complete genetic information and are bound by stringent privacy regulations that restrict data sharing and accessibility (25).

Consequently, bed/bigwig formats have become the prevailing method of representing DNA methylation data. These formats store the average methylation levels of individual CpG sites or the counts of methylated and unmethylated CpGs. Although these formats are compact and appropriate for array-based data, they sacrifice fragment-level information, including the binary methylation patterns of neighboring CpGs within the same DNA fragment. The interdependencies among CpGs within a single fragment encompass crucial aspects of DNA methylation–based analysis, particularly for tissue-of-origin inference, in both health and disease (4, 5, 6, 7, 26).

*wgbstools* offers a solution for converting DNA methylation data from BAM format into the compact and indexed PAT format (Fig 1),

including long-read sequencing technologies, such as from Oxford Nanopore (Fig S1) (27). This conversion involves merging read pairs, discarding all non-CpG positions, sorting by genomic position, collating fragments that cover the same CpG sites with identical methylation patterns, and indexing the data at the fragment level, resulting in a methylation-lossless compressed format that is typically ~300-fold smaller than the input BAM files (Table S1). The PAT format is founded on principles similar to those of the epiread format introduced by *DNMTools* (23) and *biscuit* (28), yet it is distinguished by a complementary set of utilities. These utilities encompass visualization, clipping, aggregation, filtering by the number of covered CpGs or using epigenetic fragment-level data and/or genetic information, and other commonly employed features (Table S2).

*wgbstools* also supports the CpG-centered binary format BETA, which stores the total number of reads covering each CpG, and how often it was methylated (Fig 1). Files of this format use a fixed compact size of ~50 Mb, allowing efficient direct access for target CpGs, simultaneously done across hundreds of samples.

### Analysis and visualization

*wgbstools* provides extensive support for data manipulation and visualization, both for command-line explorations and for figure generation. Users can efficiently perform tasks such as read subsampling for simulating different sequencing depths, merging

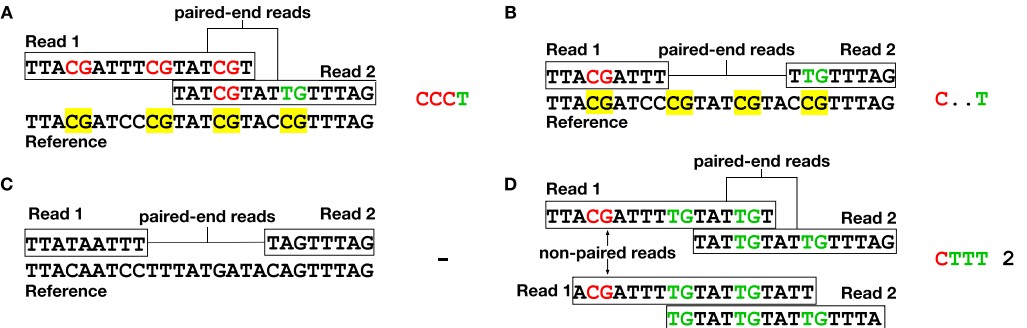

**Figure 2. Interpretation of methylated DNA sequences. bam2pat** infers methylation state for each sequenced CpG site by comparing sequenced reads with reference genome, and merges overlapping reads to produce the binary methylation pattern (PAT format). **(A)** Overlapping read pairs (i.e., read pairs marked in paired-end sequencing as coming from the same molecule), aligned against the reference genome (CpG sites marked in yellow). After bisulfite or enzymatic conversion, all cytosines are converted and read as T, whereas methylated cytosines are protected and read as C. Shown are three methylated CpGs, followed by an unmethylated one (denoted by "CCCT" in the PAT file). **(B)** Same as (A), for a nonoverlapping read pair. The methylation status of the two center CpGs is unknown and denoted as a "C..T" pattern. **(C)** Sequenced read pair that overlaps no CpG sites, and is discarded from the PAT file. **(D)** DNA fragments, showing the same binary pattern across the same CpG sites, are merged (and counted) for compactness (e.g., "CTTT 2").

replicates, or simulating in silico admixtures by combining files at various concentrations.

We have also developed a statistical framework that enables the identification of genomic regions exhibiting bimodal methylation patterns, by which half of the sequenced fragments are typically methylated, whereas the other half are mostly unmethylated, for example, in regions of allele-specific methylation because of parental imprinting or meQTL genetic variation. These regions can be further explored using the *wgbstools* **split_by_meth** and **split_by_allele** features, which select specific reads from an input BAM file, based on their fragment-level methylation or genotype at a target single nucleotide polymorphism (SNP), thus facilitating analysis of allele-specific methylation and parental imprinting (listed below).

*wgbstools* offers a range of informative visualizations at both fragment-level and single-CpG resolutions, enabling the examination of binary patterns and average methylation across multiple files. Overall, *wgbstools* presents a comprehensive command-line suite that encompasses data processing, analysis, anonymization, visualization, and manipulation tasks. It can be used independently or in conjunction with other tools such as the UCSC genome browser (29), IGV (30), or biscuit (28) for enhanced functionality.

### .pat and .beta—fragment-level and CpG-oriented data formats

BAM file information is analyzed and converted into two data formats, at complementary levels of representation and compactness. The first format, called PAT, preserves the fragment-level DNA methylation data via the **bam2pat** command. First, the methylation sites in the reference genome are indexed as $CpG_1$, $CpG_2$, $CpG_3$, …, $CpG_N$ with N = 28,217,448 for hg19. Each sequenced read is then projected onto this N-dimensional space, by discarding non-CpG positions. To further conserve space, sequenced reads (or read pairs) that cover the same set of CpG sites and show an overall identical methylation pattern are merged and counted together. For paired-end sequencing, two read pairs with the same read ID are counted as one methylation pattern (Fig 2A–C). Overall, the CpG status of each read, or read pair in the case of paired-end sequencing, appears exactly one time in a PAT file. Two methylation patterns that are identical, however coming from different

fragments/read IDs, are shown in one line but with increased count (Fig 2D). PAT files are compressed using *bgzip* and indexed for rapid direct access using *tabix* (31). Common SNPs often fall on CpG sites and hinder the possibility of proper methylation calling; thus, the **mask_pat** provides the functionality to provide a bed file of loci, which will be removed from analysis. Other common genomic annotations to include in the blacklist are low mappability regions and sex chromosomes.

The second format, called BETA, retains methylation counts for individual CpGs. These files are of fixed size, equivalent to the size of the reference genome. For each CpG, two numbers are stored as unsigned integers (8/16 bits): the count of methylated observations and the total count of observations (Fig 1). This format only keeps CpG information (as opposed to fragment-level information which is kept in PAT files), to prioritize compression efficiency and runtime. The beta files enable ultrafast reading, writing, and direct access, making them particularly advantageous for aggregating and integrating information from a large number of samples.

Below, we describe the main features of *wgbstools*.

### The view command
The **view** command allows users to retrieve and display all fragments from a PAT file, within a specified genomic region, within a set of regions, or across the entire genome. The command also provides various filtering options, including fragment length (i.e., number of covered CpGs), fragment clipping to specific boundaries, fragment subsampling. When applied to a BETA file, the **view** command displays the per-CpG binary methylation information, as well as chromosomal coordinates, in BED format (29). Overall, the **view** command provides a flexible and customizable approach for viewing and filtering methylation data, enabling users to extract specific subsets of data for further analysis and exploration.

### The vis and pat_fig commands
The **vis** command offers in-terminal visualizations for one or more samples in parallel, focusing on a target genomic region. This command uses ANSI colors and symbols for clear, lightweight, visually compelling graphics, printed directly to the terminal for interactive explorations (Fig 3).

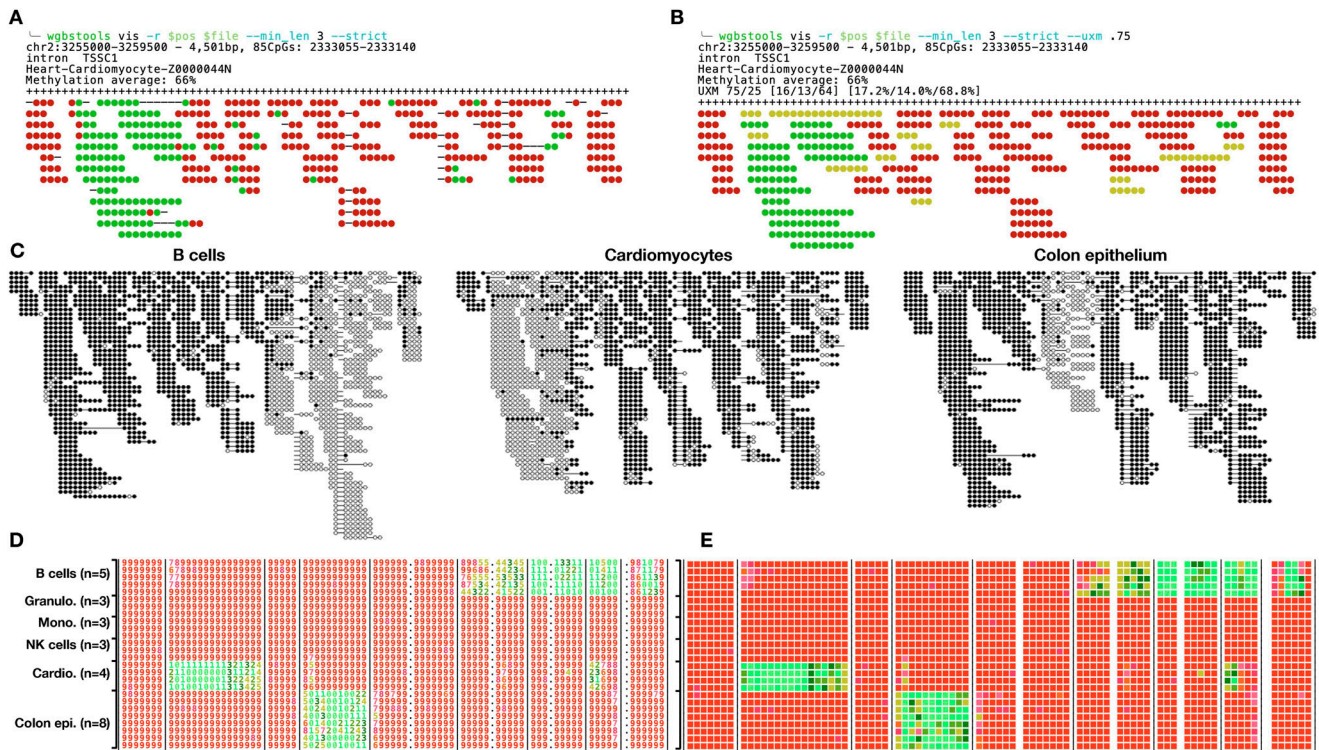

**Figure 3. *wgbstools*' vis command.**
**(A)** Fragment-level command-line interface for the visualization of a PAT file. Each individual string shows one sequenced fragment, whereas red circles denote methylated CpGs and green unmethylated ones. Strikethrough positions denote missing CpGs (e.g., in nonoverlapping paired-end reads from longer DNA fragments). Fragments are stacked and aligned by mapped positions. Shown are 85 CpGs from a 4.5-Kb region from purified cardiomyocytes. **(B)** Same as (A), where fragments are classified by their overall fragment-level methylation state (green: ≤25% methylated CpGs; red: ≥75%; otherwise yellow). **(C)** PDF visualization of multiple PAT files, for the same genomic region, including cardiomyocytes, colon epithelial cells, B cells, all from reference 5. **(D)** Screenshot of command-line interface, showing average methylation per CpG/sample in 0–9 range, in color using BETA files as input. **(E)** Same as (D), using the --heatmap flag, and the --blocks option, which uses vertical bars based on genomic segmentation of homogeneously methylated regions.

When applied to PAT files, the **vis** command displays a stacked view of aligned fragments from this target region. Each fragment is visualized as a continuous string of CpGs, colored based on their methylation. Optional features include the ability to filter reads by length, clip reads to the specific region, classify and color reads based on their overall average methylation, randomly shuffle read order, print genomic boundaries, and print regional statistics and annotations.

Similarly, **pat_fig** generates high-quality publication-ready plots of methylation data from PAT files, in a variety of file formats (e.g., pdf, png), as well as multiple esthetic customizations (plot size, color schemes, spacing, etc.; see Fig 3).

When applied to BETA files, the **vis** command computes the average methylation for each CpG in each sample and prints a colorful heatmap. For compactness, average methylation values are discretized into values ranging from 0 to 9, represented by a single-colored ASCII character. This approach allows an informative and visually appealing visualization of the methylation patterns in the selected CpG sites.

As shown in Fig 3, the **vis** command provides a convenient and visually intuitive way to explore and analyze methylation data directly in the terminal, enabling users to gain valuable insights and make informed interpretations of their data.

### The segment and beta_to_table commands

Neighboring CpGs often display correlated methylation patterns in a block-like fashion (4, 5, 6). The **segment** uses a computational, dynamic programming algorithm, to find the optimal segmentation of the genome into contiguous regions of neighboring CpGs that show similar methylation levels across samples (Fig 3E) (5). The output of this command is a BED file, where the genomic coordinates and CpG IDs of each block are denoted. Parameters include a Bayesian pseudocount value, which controls the trade-off between block length and homogeneity; a list of BETA files on which the (concurrent multi-channel) segmentation is performed; and thresholds for minimal and maximal block sizes (in bases or CpGs).

Once a block file is obtained, the **beta_to_table** command processes a set of samples (in BETA format) and produces a tab-separated table showing the average methylation of each region (row) across each sample (column). Given an additional "groups" file, the command allows aggregation over replicates or samples of the same cell type. To reduce noise, the program can treat low-coverage regions, where few reads are present, as missing values.

**beta_to_table** is applicable to any BED file, including custom cell type–specific markers, differentially methylated regions, or

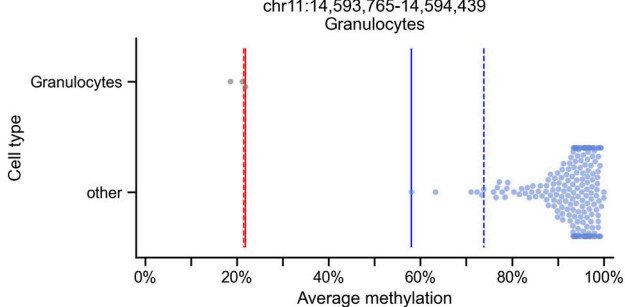

**Figure 4. Marker finding specification.**
Shown are the average methylation values of 205 WGBS samples from reference 5 in a block of 5 CpGs (chr11:14,593,765-14,594,439, 674 bp, hg19). **find_markers** may identify this block as a marker for granulocyte cells as the "target group" because the methylation values are low in the granulocyte samples (black circles, n = 3) and high in the other samples (light blue circles, n = 202). **find_markers** tests the values of, and the difference between, the percentiles specified by the user, in this case, the 2.5th percentile of the background group (dashed blue line) and the 75th percentile of the target group (dashed red), as well as the difference between the minimum of the background group (solid blue) and the maximum of the target group (solid red). The $t$ test $P$-value ($2.8 \times 10^{-46}$) is reported and is also used by **find_markers** for block filtering.

other regions of interest in the genome. This segmentation and block representation allow an easy dimensionality reduction, moving from the 28 million CpG space to a compact biologically meaningful representation of whole-genome DNA methylation.

### The find_markers command

The **find_markers** command identifies differentially methylated regions (DMRs) that exhibit unique methylation patterns in one or more samples from a specific cell type or condition (versus all or some others). For instance, in a dataset consisting of k distinct cell types, each in multiple replicates, the command identifies and ranks the top genomic regions (blocks) that are specifically unmethylated in most replicates of one cell type, compared with the samples in the remaining k-1 cell types (one-versus-all approach).

**find_markers** requires users to input an initial set of blocks (e.g., a capture-panel design file, a set of promoters or other regions of interest, or regions found using the **segment** command), as well as a collection of samples to be compared (BETA format), and a "groups" file assigning each sample to its respective group/cell type. Markers are scored and ranked using several possible scores, and filtered by statistical significance (Fig 4; see the Materials and Methods section) (5). The percentiles used to remove outliers in the target and background groups are controlled by the parameters "--tg_quant" and "--bg_quant."

### The pat2beta command

The **pat2beta** command is designed to process a pat file and generate a corresponding beta file. The resulting beta file stores, for each CpG site, the number of reads covering the site and the number of observed methylated instances of the site (Fig 1). This command creates a compact representation of marginal methylation data in the beta file format.

### The convert command

The **convert** command provides functionality for translating genomic loci between CpG indexes and genomic coordinates. In *wgbstools*, genomic regions are indexed using CpG-based coordinates, denoted as $CpG_1$, $CpG_2$, $CpG_3$, and so on. This indexing approach optimizes memory usage and running time. The **convert** command takes a bed file as input and adds columns that indicate the CpGs covered by each line in the bed file. Moreover, it appends genomic annotations for each region, providing valuable context and information for the converted data.

### The homog command

This command is designed for fragment-level analysis of next-generation sequencing DNA methylation data. Given a PAT file and an input BED file containing a set of regions of interest, **homog** classifies each sequenced fragment as mostly unmethylated (U), mostly methylated (M), or mixed (X), and outputs for each genomic regions the number U, X, and M fragments. The BED file must include at least five columns: chromosome, start, end, start-CpG-index, and end-CpG-index, as outputted by the **convert** command. By default, **homog** considers fragments covering ≥3 CpGs, and defines U fragments as those having ≤25% methylated CpG sites, and M fragments as those with ≥75% methylated CpGs. For efficiency and compactness, the default output file is in a binary uint8 format, although tab-delimited text output is also available.

### The test_bimodal command

This command implements a statistical test to estimate whether the sequenced fragments in a given target region are likely to have been derived from a single methylation pattern (i.e., biallelic pattern), or from a two-component mixture model, representing differential, allele-specific methylation. The null hypothesis $H_0$ assumes each of the CpGs in the target region is independent of its neighboring sites, and a specific Bernoulli distribution parameter is fitted using a maximum-likelihood estimator for each CpG site. Conversely, the $H_1$ hypothesis allows two such distributions, each occurring with a prior probability of 50% (see the Materials and Methods section, Figs 5 and S2).

We tested the test_bimodal function on known imprinting control regions (32) on the data of reference 5. All of the known parentally methylated allele-specific methylation regions were identified as bimodal ($P$-value < 0.005) in at least one cell type (11).

### The init_genome command

This command is for creating all of the necessary utility files for a specific reference genome. If the reference genome name (e.g., hg19, mm9, and hg38) are provided in UCSC, then providing the genome name will cause this command to download the specified reference genome fasta. Otherwise, a custom fasta file can be used and all necessary reference support files will be generated from this file.

### The mask_pat command

This command takes as input a bed file and masks all single CpGs in a pat file that intersect the regions in the input file. The resulting pat file will show a missing/masked "." value at these CpG sites. This is useful in bisulfite sequencing analysis to mask CpG sites

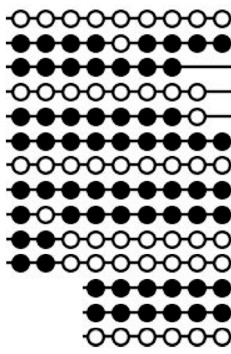

### Cardiomyocytes
chr2:3255336-3255909

### Cardiomyocytes
chr11:2023385-2023483

Avg. Meth.

Avg. Meth. Allele 1

Avg. Meth. Allele 2

**p-val: 1.0**
Single-allele LL: -109.34
Two-allele LL: -138.79

**p-val: <1e-320**
Single-allele LL: -782.93
Two-allele LL: -242.5

**Figure 5.  test_bimodal function identifies bimodal regions with significant *P*-values.** We show methylation of CpG sites on sequenced fragments from a single cardiomyocyte sample at two distinct regions. The first region, on the left, is mostly methylated and then becomes mostly unmethylated; however, each CpG site has the same methylation pattern across all sequenced fragments, and is thus not bimodal and receives an insignificant *P*-value. The second region, on the right, is from a known imprinting control region with allele-specific methylation. A probabilistic model allowing two distinct alleles each with differing probabilities for each CpG site to be methylated is more likely than a single-allele model.

where polymorphisms can obfuscate methylation calling and provide undesired genetic information, as well as to mask regions to be excluded from analysis.

## Discussion

The compact nature of the formats generated by *wgbstools* significantly reduces the disk space and memory requirements for analyzing DNA methylation sequencing data. This is particularly crucial in the face of the escalating volume of data generated by next-generation sequencing technologies and the increasing popularity of DNA methylation studies, allowing for joint analysis and interpretation of multiple samples. The ease of downstream analysis is thereby enhanced, allowing researchers to handle larger datasets more efficiently, and to foster the exploration of complex relationships within methylation patterns. The *wgbstools* PAT format is used by the UXM deconvolution method (5), a fragment-level reference-based computational algorithm for DNA methylation sequencing data, that is useful for identifying cell-of-origin in liquid biopsy and computational pathology adopted by several recent works (33, 34, 35).

*wgbstools* introduces a paradigm shift in the sharing of methylation sequencing data. Today, researchers commonly release derivative formats (beds/bigWigs) of their data publicly, while withholding or restricting access to the raw sequencing data because of its sensitive nature containing private information about donors. In contrast, our presented PAT format achieves a delicate balance, revealing only the methylation pattern, while discarding the DNA sequence, which holds the more sensitive information. This anonymization proves to be a significant asset,

particularly for large-scale human methylation projects like Roadmap Epigenomics (36), BLUEPRINT (37), and the comprehensive human WGBS atlas by reference 5. By providing an ultracompact CpG-oriented representation of high-throughput data, researchers can now share methylation data without compromising privacy, facilitating broader collaboration and accelerating scientific discoveries.

Although *wgbstools* excels in CpG-specific resolution, it does come with a trade-off. The tool prioritizes CpG sites, potentially neglecting valuable information such as motifs or fragment start/ends crucial for fragmentomic analysis. Researchers interested in retaining such information while leveraging some of *wgbstools* features could use *wgbstools*' **add_cpg_counts** feature, which adds the methylation pattern (e.g., CCCTCTCTT) as an additional tag for each read pair in the BAM file, allowing users to avoid fully transitioning to the PAT format domain.

Another advantage of *wgbstools* is that it works for analyzing other cytosine conversion–based sequencing data like TAPS-seq (modC conversion) and ACE-seq (5hmC conversion). Currently, ternary methylation codes for modC and 5hmC data, where T would mark unmodified cytosines, C would mark 5mCpGs, and H would mark 5hmCpGs, as measured using Biomodal's 6-letter sequencing, are currently not supported and are planned for upcoming future work and releases.

*wgbstools* offers universal compatibility by supporting any BAM file, irrespective of the sequencing technology employed, be it Illumina, Ultima Genomics, Oxford Nanopore, or others. This flexibility ensures that researchers can seamlessly integrate data from diverse sources, promoting cross-platform comparisons and enhancing the robustness of methylation analyses.

As sequencing-based DNA methylation data analysis is gaining popularity and is rapidly evolving, there is a growing need for

software for data processing, representation, visualization, and analysis. *wgbstools* is a flexible suite that allows researchers from the biomedical community to analyze and visualize DNA methylation sequencing data efficiently and with ease, and will pave the way for novel discoveries, including cell type–specific enhancers, novel imprinting regions, and novel methylation-based biomarkers for circulating cell-free DNA.

# Materials and Methods

### init_genome reference file creation

When the **init_genome** command is run with hg38, hg19, or other genomes, which exist in the UCSC database, then they are downloaded from https://hgdownload.soe.ucsc.edu/goldenPath/[genome]/bigZips/[genome_name].fa.gz. Any reference location with a cytosine followed by a guanine is labeled as a CpG site. The user may run init_genome with their own fasta file.

### bam2pat usage

The **bam2pat** function uses SAMtools as a preprocessing step. This allows filtering reads based on SAMtools flags. All parameters are configurable by the user; however, here we list the default values. The defaults are -f 3 (only if file is paired-end), -F 1796, and -q 10. To run on long-read sequencing, such as ONT and PacBio, data use the "-np" flag.

### Differentially methylated block identification method

The **find_markers** command essentially filters a set of blocks to a subset of differentially methylated ones, by selecting blocks where the average methylation value of the samples in one set ("target" group) is different from the values in the other set of samples ("background" group). Multiple statistics are computed for each block and each group, such as the average of averages, the $k^{th}$ percentile, minimum/maximum, and the percentage of missing values in each group. It then filters the blocks by multiple conditions, including differences in the statistics between the two groups, absolute thresholds of the statistics, significance level of a statistical test (configurable by the user), direction (i.e., methylated versus unmethylated), block size, and number of CpGs. This allows the user to filter and sort the markers by these statistics, the differences between them, and a *P*-value (see Fig 4 for an example). *wgbstools* calculates *P*-values for a *t* test and Mann–Whitney *U* test comparing the methylation levels of the target and background, as well as Welch's *t* test comparing the $\log_2$-transformed average methylated-to-average unmethylated CpG ratio. This calculation is an extension of M-values (38, 39) for regions covering multiple CpGs.

### test_bimodal

We calculate a *P*-value for the significance of a two-allele model being more likely than a single-allele model for a specific region. We assume that each CpG site has its own probability of being methylated. That is, for CpG site *i* the probability to be methylated is $\theta_i$, and any read intersecting this site has probability $\theta_i$ to be methylated at that CpG site. Given the probability of being methylated, we assume the methylation status of each CpG site in each read is independent of the status at other sites. If we let $r^j$ be read *j* and $r_i^j$ be the methylation status of read *j* at the CpG site *i*, it equals 1 if it is methylated and 0 otherwise. For the single-allele model, the likelihood can then be written as $\prod_j \prod_i \theta_i^{r_i^j}(1 - \theta_i)^{1-r_i^j}$. We use the maximum-likelihood estimator for the probability to be methylated at each CpG site (# methylated observations/total observations).

For the two-allele model, we let A be a variable representing whether a read is distributed according to allele 1 or allele 2. Thus, the likelihood can be written as follows:

$$\prod_j P\left(r^j\right) = \prod_j P\left(r^j | A = 1\right) P(A = 1) + P\left(r^j | A = 2\right) P(A = 2)$$

$$= \prod_j \prod_i \frac{1}{2}\left(\theta_j^1\right)^{r_i^j}\left(1 - \theta_j^1\right)^{1-r_i^j} + \frac{1}{2}\left(\theta_i^2\right)^{r_i^j}\left(1 - \theta_i^2\right)^{1-r_i^j}$$

This can be seen in Fig S1. We use an expectation–maximization (EM) algorithm to calculate the likelihood by iteratively associating reads with an allele and estimate the probability to be methylated at each CpG site for each allele. Finally, we calculate a *P*-value using the log-likelihood ratio test to ensure that the two-allele model significantly describes the data better than the one-allele model.

### System requirements

*wgbstools* functions exclusively on Unix-based systems and is not compatible with Windows. Leveraging standard C++ libraries, widely used Python packages, and established bioinformatics software such as SAMtools (24), *wgbstools* is easy to install. Specific features may necessitate additional Python packages; users can find detailed requirements on the project's GitHub page.

# Data Availability

The source code is available at https://github.com/nloyfer/wgbs_tools. This code is distributed under the Hebrew University of Jerusalem open software research license. The license agreement is available at https://github.com/nloyfer/wgbs_tools/blob/master/LICENSE.md. WGBS samples were downloaded from GSE186458.

# Supplementary Information

# Acknowledgements

This work was supported by the Israel Science Foundation (grants no. 1250/18, 3099/22, 259/23), by Horizon Europe (PANCAID consortium), by the Israeli

Ministry of Science and Technology (a knowledge center for forensic DNA), Israel Innovation Authority (LiquidBx consortium), and by the Center for Interdisciplinary Data Science Research at the Hebrew University.

## Author Contributions

N Loyfer: conceptualization, software, formal analysis, investigation, visualization, methodology, and writing—original draft, review, and editing.
J Rosenski: conceptualization, software, formal analysis, investigation, visualization, methodology, and writing—original draft, review, and editing.
T Kaplan: conceptualization, formal analysis, supervision, investigation, methodology, and writing—original draft, review, and editing.

## Conflict of Interest Statement

The authors declare that they have no conflict of interest.

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
