## [Reviewer comments · Life Science Alliance]

wgbstools: A computational suite for DNA methylation sequencing data analysis

Netanel Loyfer, Jonathan Rosenski, and Tommy Kaplan

DOI: <https://doi.org/10.26508/lsa.202503514>

Corresponding author(s): Tommy Kaplan, Hebrew University of Jerusalem

Review Timeline:

Submission Date:	2025-09-21
Editorial Decision:	2025-10-23
Revision Received:	2025-12-15
Editorial Decision:	2026-01-06
Revision Received:	2026-01-18
Accepted:	2026-01-20

Scientific Editor: Tim Fessenden

Transaction Report:

October 23, 2025

Re: Life Science Alliance manuscript #LSA-2025-03514-T

Jonathan Rosenski

Dear Dr. Rosenski,

Thank you for submitting your manuscript entitled "wgbstools: A computational suite for DNA methylation sequencing data representation, visualization, and analysis" to Life Science Alliance. The manuscript was assessed by expert reviewers, whose comments are appended to this letter. We invite you to submit a revised manuscript addressing the reviewer comments.

As you will see, reviewers were unanimous in their enthusiasm for the wgbstools package for handling DNA methylation sequencing data documented in this work. Reviewer 2 pointed to a potential source of bias that should be addressed or clarified in the text. Reviewer 3 sought additional details on the construction of CpGs from the hg19 build. Reviewers 1 and 3 made additional suggestions to improve this work which we invite you to consider.

Thank you for this interesting contribution to Life Science Alliance. We are looking forward to receiving your revised manuscript.

Sincerely,

B. MANUSCRIPT ORGANIZATION AND FORMATTING:

Reviewer #1 (Comments to the Authors (Required)):

This is a software resource and method paper that describes wgbstools, a software toolkit for DNA methylation sequencing data. We have been early adopters of wgbstools since it was first published as a preprint in 2024, and it has been a very helpful tool. A key feature of the software is the ability to handle single-molecule phased fragment data and to compress this data >100X. The useful data format it introduces is the PAT file, which retains the valuable fragment-level or phased methylation data from sequencing. In our experience, there has been no reason to realign large-scale original FASTQ data if PAT files are available, and we trust the handling of the original data. The PAT format is much faster to implement and can be done on a laptop as opposed to realignment on a large server over the course of weeks. Such improvement is non-trivial, and the software has been robust across different platforms. Overall, I believe this will be a valuable resource to the community and timely in light of the shift away from methylation arrays and towards methylation sequencing datasets. I only have minor suggestions.

Minor:

1) Removing genetic identifiers from aligned BAM files has tremendous practical utility in sharing epigenetic clinical datasets. However, some common genotypes overlap CpG positions, as the C to T mutation is common, making re-identification possible even with CpGs. Could the authors provide a way to remove such positions if a BED file of common SNP positions were provided?

2) Can the authors show an example wgbstools dataset with long-read sequencing data? This can be done with publicly available data. This is an important point, as in addition to Oxford Nanopore and Pacbio, there is now Roche's upcoming SBX technology and Illumina's constellation technology that may enable phasing of both methylation and genotypes across long stretches of the genome. This would prove helpful in generating informative data for discovery.

3) When using the "bam2pat" command for data conversion, were the sequencing reads screened before that? Such as mapping quality, etc. If so, please list the specific parameters and values and explain the reason for them.

4) When using "find_markers" to screen for DMRs, can you comment on the default settings of 2.5th percentile of the background group and the 75th percentile of the target group as the?

- 5) Find_markers uses a t-test to set a cut-off. Can the authors comment on the normality and homoscedasticity of typical datasets? Why not use non-parametric statistics?
- 6) Figure 3C's caption describes "BETA" files, but we see fragment-level files in the figure that likely come from the PAT format?
- 7) A key feature is the degree of compression, which is seen in Table S1. This seems like a key feature worthy of mentioning in the abstract.

Reviewer #2 (Comments to the Authors (Required)):

Loyfer et al. present wgbstools, a suite of software for the processing, analysis, and visualisation of whole-genome bisulphite sequencing data. The programme is presented in detail, and explanations and justifications for the various functions are detailed and adequate, respectively. Prior to publication, there is only one possible issue which I would like to see the authors address:

When merging overlapping (but not identical) fragments, it appears that the overlapping CpG sites are not counted, and that this only happens if there are two identical fragments. Can the authors please confirm if this is correct?

If the above is correct, then bias could be introduced into the data in cases where a CpG site is in an overlapping region of two non-identical strands. This would lead to double counting of the CpG site, but it would not be annotated as having been double-counted in the PAT file format. A possible solution to this could be to indicate the number of times each CpG site was counted in merged fragments in the PAT file. To take the example used in Figure 2D, the PAM format could be CTTT 2332 (rather than CTTT 2) to indicate the count of each CpG site. To extend my example, the PAM format of Figure 2A would be CCCT 1121.

Reviewer #3 (Comments to the Authors (Required)):

Loyfer et al have produced a flexible Unix tool for human CpG methylome data - wgbstools.

This has been highly useful for the authors having employed this in their DNAm atlas as well as other papers. It incorporates standard python packages, C++ libraries and SAMtools. My only concern is its applicability moving forward. My comments for the authors to consider are listed below

Major

1. Could the authors include details about how the set of ~28.2M hg19 CpGs was constructed? Some analyses only include autosomal CpGs to avoid issues such as random X inactivation etc - and can adjust for population-specific variation.
[https://www.biocductor.org/packages/devel/bioc/vignettes/ramwas/inst/doc/RW2_CpG_sets.html#:~:text=Our%20CpG%20sets%20include%20all,silico%20experiment%20\(details%20below\)](https://www.biocductor.org/packages/devel/bioc/vignettes/ramwas/inst/doc/RW2_CpG_sets.html#:~:text=Our%20CpG%20sets%20include%20all,silico%20experiment%20(details%20below)).
2. Authors have used the old human genome build hg19 as reference - are files for hg38 also available as standard to increase the usability of the package?
3. Additionally, the T2T human genome has an estimated ~32M CpGs now - so could also consider including this set for the additional benefits of this alignment.
4. Do the authors see this package as adaptable for long read data, e.g., incorporating 5hmC - or will this require a different approach/new package?

Minor

1. In the Abstract - instead of 'increased coverage of methylation sites by 30-60x' - which could be misinterpreted on a quick read as discussing sequencing depth - the authors could rewrite this sentence for clarity.
2. In Introduction include EPIC v1 and v2 - and increased percentage CpG inclusion of the latter.
3. Error in Ziller et al - reference on pg 3 '8/2013'
4. Song et al. reference for DNMTTools is missing

Point-by-point response for: wgbstools: A computational suite for DNA methylation sequencing data representation, visualization, and analysis

Reviewer #1

This is a software resource and method paper that describes wgbstools, a software toolkit for DNA methylation sequencing data. We have been early adopters of wgbstools since it was first published as a preprint in 2024, and it has been a very helpful tool. A key feature of the software is the ability to handle single-molecule phased fragment data and to compress this data >100X. The useful data format it introduces is the PAT file, which retains the valuable fragment-level or phased methylation data from sequencing. In our experience, there has been no reason to realign large-scale original FASTQ data if PAT files are available, and we trust the handling of the original data. The PAT format is much faster to implement and can be done on a laptop as opposed to realignment on a large server over the course of weeks. Such improvement is non-trivial, and the software has been robust across different platforms.

Overall, I believe this will be a valuable resource to the community and timely in light of the shift away from methylation arrays and towards methylation sequencing datasets. I only have minor suggestions.

Minor:

1) Removing genetic identifiers from aligned BAM files has tremendous practical utility in sharing epigenetic clinical datasets. However, some common genotypes overlap CpG positions, as the C to T mutation is common, making re-identification possible even with CpGs. Could the authors provide a way to remove such positions if a BED file of common SNP positions were provided?

We agree that potential re-identification via SNPs is a critical concern for clinical data sharing. To address this, we have introduced into wgbstools a specific command, `mask_pat`, which masks CpG sites intersecting with an input BED file of common SNPs. Another common usage is the masking of "blacklisted" regions, with dubious mapping/alignability (e.g. from ENCODE). We have now added to the Methods section a detailed explanation of this feature and its applications, to ensure users can easily anonymize their datasets when needed.

2) Can the authors show an example wgbstools dataset with long-read sequencing data? This can be done with publicly available data. This is an important point, as in addition to Oxford Nanopore and Pacbio, there is now Roche's upcoming SBX technology and Illumina's constellation technology that may enable phasing of both methylation and genotypes across long stretches of the genome. This would prove helpful in generating informative data for discovery.

We thank the reviewer for this opportunity to demonstrate its utility with long-read data. Indeed, wgbstools supports Nanopore (ONT) and PacBio data via the `bam2pat` "-np" flag. We have now

added Figure S2, which utilizes publicly available ONT data (sample HG001) to visualize methylation of reads covering >100 CpGs, a scale achievable only with long reads. We have also updated the Methods section to include specific instructions for processing long-read alignments.

3) When using the "bam2pat" command for data conversion, were the sequencing reads screened before that? Such as mapping quality, etc. If so, please list the specific parameters and values and explain the reason for them.

There is no need for pre-screening. The required SAM flags for inclusion/filtering are included in the "bam2pat" command by default, and the user could further change them. For paired-end sequencing data, an inclusion parameter of "-f 3" is used (read is paired and mapped properly in the pair) by default. The exclusion flag, by default is "-F 1796" thus ignoring reads that are not mapped, not in the primary alignment, fail platform/vendor quality checks, or are PCR or optical duplicates. For mapping quality the value of 10 is used, however this is configurable. We have noted the default samtools flags used in the methods section, as well as the "bam2pat" flags used for modifying these options ("--include_flags", "--exclude_flags", "--mapq").

4) When using "find_markers" to screen for DMRs, can you comment on the default settings of 2.5th percentile of the background group and the 75th percentile of the target group as the?

Indeed, these default parameters were empirically derived from our goals of identifying tissue-specific markers of our previously published large-scale methylation atlas (e.g., ~200 background samples vs. ~3 target samples). We aimed to be permissive of limited outliers: allowing ~25% outliers in the small target group (e.g., 1 sample) and ~2.5% in the (typically much larger) background set. This ensures that users processing similar atlas-scale datasets can reproduce established marker sets. Obviously, these parameters could be controlled by "find_markers" flags ("--tg_quant" and "--bg_quant"), as we now specify in the manuscript.

5) Find_markers uses a t-test to set a cut-off. Can the authors comment on the normality and homoscedasticity of typical datasets? Why not use non-parametric statistics?

This is an excellent point, as methylation beta-values often violate assumptions of normality and homoscedasticity. To address this, we have now implemented two robust alternatives to the standard t-test: the non-parametric Mann-Whitney U test and Welch's t-test applied after an M-

value transformation (Logit), which stabilizes variance (Ladd-Acosta et al. 2010; Du et al. 2010). These options are now available in the software and documented in the Methods section.

Thanks!

6) Figure 3C's caption describes "BETA" files, but we see fragment-level files in the figure that likely come from the PAT format?

You are correct, thank you. This is fixed now.

7) A key feature is the degree of compression, which is seen in Table S1. This seems like a key feature worthy of mentioning in the abstract.

Done.

Reviewer #2

Loyfer et al. present wgbstools, a suite of software for the processing, analysis, and visualisation of whole-genome bisulphite sequencing data. The programme is presented in detail, and explanations and justifications for the various functions are detailed and adequate, respectively. Prior to publication, there is only one possible issue which I would like to see the authors address:

When merging overlapping (but not identical) fragments, it appears that the overlapping CpG sites are not counted, and that this only happens if there are two identical fragments. Can the authors please confirm if this is correct?

We apologize if the text was not clear enough. In paired-end sequencing, we begin by identifying the two reads (same read ID), and using the BAM CIGAR string to properly align them. It should be noted that these do NOT represent the top and bottom strands of the original fragment. Instead, both orientations were synthesized (using “bridge” amplification, from a single-stranded DNA fragment), as shown in Figure 3, here:

https://www.illumina.com/documents/products/techspotlights/techspotlight_sequencing.pdf

We therefore merge the two reads, to produce their joint binary pattern, for all covered CpGs. CpGs that fall between the two reads, and are uncovered, are marked by dots (as we show in Figure 2B). In the case where a CpG is covered by both read1 and read2, but shows contradicting values (due to a sequencing error), we mask that specific CpG and show a missing value.

At the end of this stage, each read pair is replaced by a string of C's, T's (and possibly . 's). This is the basis for the .pat file. To further reduce the file size, two DNA fragments (i.e. two pairs of sequenced reads) that cover the same CpGs and show the same binary pattern (as in Fig 2D), are “merged” into one line in the .pat file, but we increase the “count” of this pattern to n=2.

We have now revised the diagram in Figure 2 and the accompanying text to clearly distinguish between 'read merging' and read-pair merging' to prevent this confusion. Thanks!

If the above is correct, then bias could be introduced into the data in cases where a CpG site is in an overlapping region of two non-identical strands. This would lead to double counting of the CpG site, but it would not be annotated as having been double-counted in the PAT file format. A possible solution to this could be to indicate the number of times each CpG site was counted in merged fragments in the PAT file. To take the example used in Figure 2D, the PAM format could be CTTT 2332 (rather than CTTT 2) to indicate the count of each CpG site. To extend my example, the PAM format of Figure 2A would be CCCT 1121.

We thank the reviewer for this interesting idea, which we indeed considered in earlier stages. Nonetheless, for the reasons detailed above, we saw no practical difference between CpGs that were sequenced twice (by read1 and read2) compared to those covered by a single read. We speculate that this is because of high sequencing accuracy - the chances of a sequencing error are far smaller than the natural (biological) noise per CpG, as well as bisulfite conversion errors

(together estimated at the order of ~1%). So we eventually decided to count patterns, rather than coverage per CpG per pattern. This is now clarified in the figure as well as in the main text.

Reviewer #3

Loyfer et al have produced a flexible Unix tool for human CpG methylome data - wgbstools. This has been highly useful for the authors having employed this in their DNAm atlas as well as other papers. It incorporates standard python packages, C++ libraries and SAMtools. My only concern is its applicability moving forward. My comments for the authors to consider are listed below

Major

1. Could the authors include details about how the set of ~28.2M hg19 CpGs was constructed? Some analyses only include autosomal CpGs to avoid issues such as random X inactivation etc - and can adjust for population-specific variation.

[https://www.bioconductor.org/packages/devel/bioc/vignettes/ramwas/inst/doc/RW2_CpG_sets.html#:~:text=Our%20CpG%20sets%20include%20all,silico%20experiment%20\(details%20below\)](https://www.bioconductor.org/packages/devel/bioc/vignettes/ramwas/inst/doc/RW2_CpG_sets.html#:~:text=Our%20CpG%20sets%20include%20all,silico%20experiment%20(details%20below))).

We thank the reviewer for raising this important issue. wgbstools is designed to work on any genome. The “init_genome” command accepts any FASTA reference (hg19, hg38, T2T, or non-human genomes) and dynamically identifies all Cytosine-Guanine dinucleotides to construct the reference index. Users can also provide custom FASTA files to exclude specific chromosomes or include additional contigs. We have now clarified this flexibility in the Methods section.

Moreover, an alternative to removing chromosomes in the reference genome file is to run the “bam2pat” command with the “--blacklist” option or later use the “mask_pat command”. These new features can be used for removing problematic genomic regions, such as the X chromosome, to mask CpGs that overlap common or ancestry-related SNPs, or to exclude CpGs at ENCODE blacklisted regions. This has now been clarified in the main text.

2. Authors have used the old human genome build hg19 as reference - are files for hg38 also available as standard to increase the usability of the package?

Yes, any reference fasta file can be used. To choose a specific reference genome the “init_genome” command is used and the input is a fasta file. We have now added a section into the main text to describe the process of using a different reference genome. In the atlas paper (Loyfer et al, Nature, 2023), for example, we supplied .beta and .pat files for both hg19 and hg38 genome versions (see <https://www.ncbi.nlm.nih.gov/geo/query/acc.cgi?acc=GSE186458>).

3. Additionally, the T2T human genome has an estimated ~32M CpGs now - so could also consider including this set for the additional benefits of this alignment.

As above, any reference genome fasta file can be used. To use the T2T genome, one could run “wgbstools init_genome hs1” to download the T2T reference genome from UCSC (<https://hgdownload.soe.ucsc.edu/goldenPath/hs1/bigZips/hs1.fa.gz>), install it, and use it. We thank the reviewer for this suggestion.

4. Do the authors see this package as adaptable for long read data, e.g., incorporating 5hmC - or will this require a different approach/new package?

Yes, wgbstools supports long-read data (e.g., ONT, PacBio) via the “bam2pat” command with the “-np” flag. Regarding 5hmC specifically, the applicability depends on the library preparation method. For standard native Nanopore sequencing, the current version only supports 5mC calling and analysis. We are expected to support ternary methylation code for modC and 5hmC data, where T would mark unmodified cytosines, C would mark 5mCpGs and H would mark 5hmCpGs, as measured using Biomodal's 6-letter sequencing. For TAPS-seq (modC conversion) and ACE-seq (5hmC conversion), wgbstools works out-of-the-box, but file interpretation could be more specific. These versions are still under development, and will be released during 2026 (as special care is needed for the .beta files). Regarding ONT data, we have added Figure S2 to demonstrate the tool's long-read capabilities and clarified this distinction in the Methods.

Minor

1. In the Abstract - instead of 'increased coverage of methylation sites by 30-60x' - which could be misinterpreted on a quick read as discussing sequencing depth - the authors could rewrite this sentence for clarity.

We thank the reviewer for this clarification. We have changed the sentence to: “Next-generation methylation-aware sequencing of DNA sheds light on the fundamental role of methylation in cellular function in health and disease, increasing the number of covered CpG sites from hundreds of thousands in previous array based approaches, to tens of millions across the whole genome.”

2. In Introduction include EPIC v1 and v2 - and increased percentage CpG inclusion of the latter.

Done.

3. Error in Ziller et al - reference on pg 3 '8/2013'

Done. Thank you.

4. Song et al. reference for DNMTTools is missing

Done. Thank you.

January 6, 2026

RE: Life Science Alliance Manuscript #LSA-2025-03514-TR

Prof. Tommy Kaplan
Hebrew University of Jerusalem
Givat Ram
Jerusalem 91904
Israel

Dear Dr. Kaplan,

Thank you for submitting your revised manuscript entitled "wgbstools: A computational suite for DNA methylation sequencing data analysis". As you will see, reviewers are entirely satisfied and all recommend publication. We would be happy to publish your paper in Life Science Alliance pending final revisions necessary to meet our formatting guidelines.

- Please be sure that the authorship listing and order is correct.
- Please upload all figure files individually, including the supplementary figure files; all figure legends should only appear in the main manuscript file.
- Please add ORCID ID for corresponding author--you should have received instructions on how to do so.
- Please add the X and Bluesky handles of your host institute/organization, as well as your own and/or one of the authors, in our system.
- The titles in both the system and the manuscript file must be consistent with each other.
- It is recommended to exclude figures from the manuscript text and upload them separately.
- Please incorporate any points from the Conclusion section into the Discussion; we only allow a Discussion section.
- Please rename the "Data access" section to Data Availability.

A. FINAL FILES:

B. MANUSCRIPT ORGANIZATION AND FORMATTING:

Thank you for your attention to these final processing requirements. Please revise and format the manuscript and upload materials as soon as you are able.

Sincerely,

Reviewer #1 (Comments to the Authors (Required)):

The authors have satisfied all points raised. This tool will be a useful resource for the community, and I recommend the current manuscript for publication.
See my previous comments for a summary of the paper.

Reviewer #2 (Comments to the Authors (Required)):

Loyfer et al. have responded to my original comment satisfactorily, and have subsequently revised the manuscript to avoid possible confusion. I am happy for the article to be published in its current state.

Reviewer #3 (Comments to the Authors (Required)):

All my comments have been clearly answered.

January 20, 2026

RE: Life Science Alliance Manuscript #LSA-2025-03514-TRR

Prof. Tommy Kaplan
Hebrew University of Jerusalem
Givat Ram
Jerusalem 91904
Israel

Dear Dr. Kaplan,

Thank you for submitting your Research Article entitled "wgbstools: A computational suite for DNA methylation sequencing data analysis". It is a pleasure to let you know that your manuscript is now accepted for publication in Life Science Alliance. Congratulations on this interesting work.

Your manuscript will now progress through copyediting and proofing. Reviews, decision letters, and point-by-point responses associated with peer-review at Life Science Alliance will be published, alongside the manuscript. If you want to opt out of having the reviewer reports and your point-by-point responses displayed, please let us know immediately.

DISTRIBUTION OF MATERIALS:

Again, congratulations on a very nice paper. I hope you found the review process to be constructive and are pleased with how the manuscript was handled editorially. We look forward to future exciting submissions from your lab.

Sincerely,
